# ENHANCED MODEL-AGNOSTIC TRAINING OF DEEP TABULAR GENERATION MODELS

## ABSTRACT

Despite the active research for tabular data synthesis, state-of-the-art methods continue to face challenges arising from the complex distribution of tabular data. To this end, we claim that the difficulty can be alleviated by making the distribution simpler using Gaussian decomposition. In this paper, we propose a training method, **Ga**ussian **D**ecomposition-based **Ge**neration of **T**abular data (GADGET), which can be applied to any generative models for tabular data. The method i) decomposes the complicated distribution of tabular data into a mixture of $K$ Gaussian distributions, ii) trains one model for each decomposed Gaussian distribution aided by our proposed self-paced learning algorithm. In other words, we do not stop at utilizing a Gaussian mixture model to discover $K$ simplified distributions but utilize their surrogate density functions for designing our self-paced learning algorithm. In our experiments with 11 datasets and 8 baselines, we show that GADGET greatly improves existing tabular data synthesis methods. In particular, a score-based generative model on our GADGET training framework achieves the state-of-the-art performance in terms of sampling quality and diversity.

## 1 INTRODUCTION

Tabular data synthesis has been a long-standing research issue in machine learning (Chawla et al., 2002; He et al., 2008; Han et al., 2005; Park et al., 2018; Lee et al., 2021). With significant advancements in deep generative modeling, many tabular data synthesis methods have been proposed, e.g., CTGAN (Xu et al., 2019), TableGAN (Park et al., 2018), and STaSy (Kim et al., 2022a). However, tabular data exhibit challenging characteristics: i) some columns can have eccentric distinct distributions and moreover, ii) we need to model the joint distribution of them. Modern deep tabular models typically consider data preprocessing to be important, which involves handling categorical features and normalizing/scaling numerical ones (Xu et al., 2019). Although some approaches attempt to address the differences in distributions, modeling the data distribution has remained a challenge, as they attempt to model mixture of distributions with a single model.

In this paper, we argue that i) the complicated tabular data distribution can be simplified by breaking it down into $K$ Gaussian-like distributions and therefore, ii) training deep generative models on these $K$ Gaussian-like distributions is notably more manageable, as discussed in Sec. 2. Based on the assertion, we propose a model-agnostic training framework that can be applied to any deep learning model for tabular data generation addressing the challenges and subsequently improving model performance. To apply the method, firstly, we employ Gaussian mixture models (GMMs) to identify $K$ Gaussian distributions that best decompose the original data distribution. We then train $K$ smaller models, each specialized in a single Gaussian distribution[1]. To further reduce training complexity, we design a self-paced learning algorithm (by referring to the surrogate probability density derived from the Gaussian distributions). Self-paced learning, inspired by human learning, enables the model to control the training difficulty autonomously. Specifically, data records with high surrogate probability density are assigned higher weights in their training loss.

---

[1]Real-world tabular data is not likely to be perfectly decomposed into $K$ Gaussians. For the sake of our discussion, we assume this simplification, but our design does not rely on such clean decomposition. Our main intuition is that learning $K$ simplified distributions separately with $K$ *small* models is easier than training a *large* model for the original distribution.

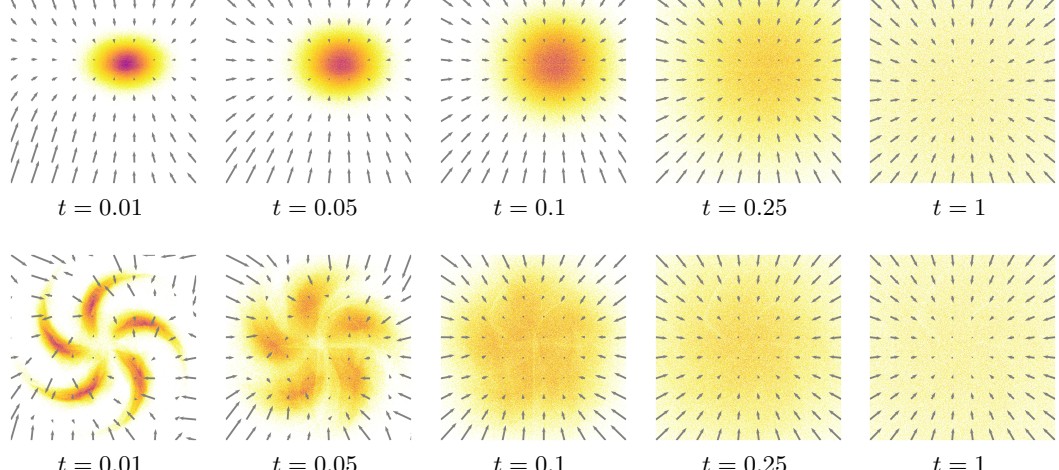

Figure 1: A vector field, i.e., *score*, and a density from trained SGM on a bivariate Gaussian distribution at time $t$ (upper) and those of a complicated distribution (lower). $t = 0$ means the original distribution, and $t = 1$ means a Gaussian prior. For our study, we design a SGM for tabular data in Appendix A but we do not claim that this is our contribution since it follows past standard designs for tabular synthesis. Hereinafter, SGMs in our contexts means tabular SGMs defined by us.

In the latter part of our paper, we demonstrate that our proposed training framework, called **Ga**ssian **D**ecomposition-based **Ge**neration of **T**abular data (GADGET), enhances the performance of deep tabular generative models. To demonstrate the effectiveness of GADGET, we thoroughly select representative generative adversarial network (GAN), flow, and score-based generative models for tabular data synthesis as base generative models of GADGET. As a result, the models trained on top of GADGET exhibit significant performance improvements. Furthermore, score-based generative models (SGMs) combined with GADGET outperforms other state-of-the-art tabular synthesis methods in terms of both sampling quality and diversity. Additionally, by decomposing the data distribution and applying our self-paced learning, we significantly reduce the number of learnable parameters in each smaller model (as well as the total number of parameters, which is the sum of the parameter numbers across small models). Our contributions are as follows:

1. We propose a model training framework GADGET which decompose the original distribution into $K$ Gaussian-like distributions. $K$ small models learn those $K$ simplified distributions, reducing the training difficulty.
2. We design a self-paced learning algorithm that utilizes the probability density of each training record for each Gaussian distribution[2].
3. We show that popular tabular models trained on the proposed framework improve the performance in terms of sampling quality, diversity, and/or time. In particular, tabular SGMs with GADGET show the best sampling quality and diversity among all methods. In addition, the number of parameters in $K$ models is mostly smaller than original models.

## 2 WHY DO WE DECOMPOSE THE ORIGINAL DISTRIBUTION?

We claim that the difficulty of modeling tabular data can be mitigated by decomposing the original distribution into $K$ Gaussian-like distributions using Gaussian mixture models (GMMs) and training $K$ small models. In the remainder of this section, we present empirical evidence illustrating the simplification of real-world datasets using Gaussian decomposition.

---

[2]In reality, it is impossible to know the exact probability density of a record $\mathbf{x}$ contained by tabular data. After the Gaussian decomposition, however, one can use the density derived from each Gaussian as a surrogate probability density measurement. Although not perfect, our experimental results prove the efficacy of the surrogate measurement since our self-placed learning based on it improves the sampling quality and diversity.

For our preliminary study, we choose SGM for its superior performance. However, our claim can be applied to other types of tabular synthesis models (see our experimental results). In Fig. 1, each figure represents a vector field, i.e., *score*, of trained SGM at time $t$. $t = 0$ means the original distribution and $t = 1$ means a Gaussian prior. The upper row is for a SGM to learn a bivariate Gaussian distribution where mean $\boldsymbol{\mu} = \left( \begin{smallmatrix} 7 \\ 7 \end{smallmatrix} \right)$ and covariance $\boldsymbol{\Sigma} = \left( \begin{smallmatrix} 4 & 0 \\ 0 & 1 \end{smallmatrix} \right)$, and the lower row is for a SGM to learn a more complicated distribution which has 5 spirals. In the lower row, the vector field becomes significantly more intricate than the upper row as $t$ approaches 0. Therefore, the training process becomes easier after Gaussian decomposition, since as the data pattern, i.e., distribution, becomes simpler, the task of model learning also becomes easier. The philosophy behind our method design can be best phrased by *solving $K$ easy problems is easier than solving one difficult problem.*

In Fig. 2, we visualize cosine similarity between vector fields with respect to time $t$ from SGMs trained for `Fish` (the upper row) and `Concrete` (the lower row). As shown, the overall similarity of SGM-GADGET is greater than that of SGM, which means the vector field of SGM-GADGET does not change drastically compared to that of

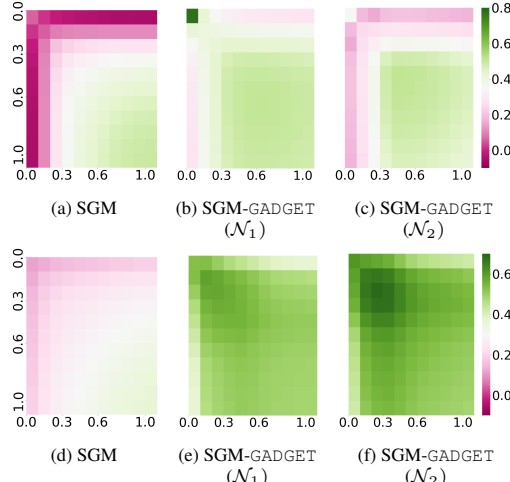

(a) SGM     (b) SGM-GADGET $(\mathcal{N}_1)$     (c) SGM-GADGET $(\mathcal{N}_2)$

(d) SGM     (e) SGM-GADGET $(\mathcal{N}_1)$     (f) SGM-GADGET $(\mathcal{N}_2)$

Figure 2: Cosine similarity between scores at each time $t$ of SGM and SGM-GADGET, where the SGM-GADGET is trained after decompose the original data distribution into 2 Gaussian-like distributions $\mathcal{N}_1$ and $\mathcal{N}_2$. The latter 2 columns represent each SGM trained on each distribution. The upper row is for `Fish`, the lower row is for `Concrete`. Each axis means time $t$, and each cell represents the averaged cosine similarity between the scores at time $t$.

SGM. Moreover, while the vector field of SGM at $t = 0$ is barely relevant to vector fields at other time $t$, the vector fields of SGM-GADGET are similar throughout all time, even at $t = 0$, i.e., the original distribution. From Fig. 2, we confirm that Gaussian decomposition using GMM works in real-world tabular data as well.

## 3    BACKGROUNDS

### 3.1    GAUSSIAN MIXTURE MODELS

The Gaussian mixture distribution is given by the following equation:

$$p(\mathbf{x}) = \sum_{k=1}^{K} \pi_k \mathcal{N}(\mathbf{x}|\boldsymbol{\mu}_k, \boldsymbol{\Sigma}_k), \tag{1}$$

where $K$ is the number of probability distributions constituting the mixture distribution, $k$-th Gaussian distribution $\mathcal{N}_k$ is characterized by a weight $\pi_k \in [0, 1]$, a mean $\boldsymbol{\mu}_k$, and a covariance $\boldsymbol{\Sigma}_k$ — we note that $\sum_k \pi_k = 1$. Eq. 1 means a linear mixture of Gaussian density functions, $\{\mathcal{N}(\mathbf{x}|\boldsymbol{\mu}_k, \boldsymbol{\Sigma}_k)\}_{k=1}^{K}$.

A mixture model is a probabilistic model for density estimation using a mixture distribution, which can describe more complex probability distributions, by combining several probability distributions. Especially, a Bayesian Gaussian mixture model is commonly extended to fit a vector of unknown parameters, or multivariate Gaussian distributions. In a multivariate distribution, one may model a vector of parameters using a Gaussian mixture model prior distribution on the vector of estimates given by Eq. 1. To incorporate this prior into a Bayesian estimation, the prior is multiplied with the known distribution $p(\mathbf{x}|\theta)$ of the data $\mathbf{x}$ conditioned on the parameters $\theta$ to be estimated. With the formulation, the posterior distribution $p(\theta|\mathbf{x})$ is also a Gaussian mixture model of the form

$$p(\theta|\mathbf{x}) = \sum_{k=1}^{K} \tilde{\pi}_k \mathcal{N}(\mathbf{x}|\tilde{\boldsymbol{\mu}}_k, \tilde{\boldsymbol{\Sigma}}_k), \tag{2}$$

where $\tilde{\pi}_k$, $\tilde{\boldsymbol{\mu}}_k$, and $\tilde{\boldsymbol{\Sigma}}_k$ for all $k$ are the updated parameters using the EM algorithm.

## 3.2 TABULAR DATA SYNTHESIS

There are many tabular data synthesis methods to create realistic fake tables for various purposes. For instance, Patki et al. (2016) utilizes a recursive table modeling with a Gaussian copula for synthesizing continuous variables. On the other hand, Bayesian networks (Zhang et al., 2017; Aviñó et al., 2018) and decision trees (Reiter, 2005) can be used for discrete variables. With great advancement in generative modeling, there exists an attempt to synthesize tabular data using GANs. TableGAN (Park et al., 2018) utilizes convolutional neural networks to improve the quality of synthesized tabular data and prediction on label column accuracy. CTGAN and TVAE (Xu et al., 2019) propose a column-type-specific preprocessing method to deal with the challenges in tabular data, for which tabular data usually consists of mixed-type variables and the variables follow multi-modal distributions. In specific, they approximate the discrete variables to the continuous spaces by using Gumbel-Softmax. OCT-GAN (Kim et al., 2021) is a generative model based on neural ODEs. SOS (Kim et al., 2022b) and STaSy (Kim et al., 2022a) are state-of-the-art tabular data synthesis methods, which are based on the score-based generative regime. The former focuses on synthesizing minority class(es) in classification data, while the latter generates the entire data. Specifically, STaSy is the most similar method to ours. In STaSy, tabular data is treated as a collection of univariate distributions, whereas SGM-GADGET handles tabular data as a single multivariate distribution. This distinction enables us to design our proposed training strategies.

## 3.3 SCORE-BASED GENERATIVE MODELS

Score-based generative models (SGMs) employ the following diffusion process of Itô stochastic differential equation (SDE):

$$\mathrm{d}\mathbf{x} = \mathbf{f}(\mathbf{x}, t)\mathrm{d}t + g(t)\mathrm{d}\mathbf{w}, \tag{3}$$

where $\mathbf{f}(\mathbf{x}, t) = f(t)\mathbf{x}$, $f$ and $g$ are drift and diffusion coefficients of $\mathbf{x}(t)$, and $\mathbf{w}$ is the standard Wiener process. We can reverse the diffusion process and this is called a denoising process:

$$d\mathbf{x} = \big(\mathbf{f}(\mathbf{x}, t) - g^2(t)\nabla_{\mathbf{x}} \log p_t(\mathbf{x})\big)dt + g(t)\mathrm{d}\mathbf{w}, \tag{4}$$

where this reverse SDE is a generative process. *score network*, the time-dependent score-based model $F_{\boldsymbol{\theta}}(\mathbf{x}, t)$, approximates the score function $\nabla_{\mathbf{x}} \log p_t(\mathbf{x})$.

We can derive $\mathbf{x}(t)$ at time $t \in [0, 1]$ using the diffusion process in Eq. 3, where $\mathbf{x}(0)$ and $\mathbf{x}(1)$ means a real and noisy sample, respectively. By this process, the transition probability $p(\mathbf{x}(t)|\mathbf{x}(0))$ at time $t$ is easily approximated, and it always follows a Gaussian distribution. It allows us to collect the gradient of the log transition probability, $\nabla_{\mathbf{x}(t)} \log p(\mathbf{x}(t)|\mathbf{x}(0))$, during the diffusion process. Therefore, we can train a score network $F_{\boldsymbol{\theta}}(\mathbf{x}, t)$ as follows:

$$\arg\min_{\boldsymbol{\theta}} \mathbb{E}_t \mathbb{E}_{\mathbf{x}(t)} \mathbb{E}_{\mathbf{x}(0)} \Big[ \lambda(t) \| F_{\boldsymbol{\theta}}(\mathbf{x}(t), t) - \nabla_{\mathbf{x}(t)} \log p(\mathbf{x}(t)|\mathbf{x}(0)) \|_2^2 \Big], \tag{5}$$

where $\lambda(t)$ is to control the trade-off between the sampling quality and likelihood. This is called *denoising score matching*, and $\boldsymbol{\theta}^*$ solving Eq. 5 can accurately solve the reverse SDE in Eq. 4 (Vincent, 2011).

After the training process, we can synthesize fake data records with i) the *predictor-corrector* framework or ii) the *probability flow* method, a deterministic method based on the ordinary differential equation (ODE) whose marginal distribution is equal to that of Eq. 3 (Song et al., 2021). In particular, the *probability flow* enables fast sampling and exact log-probability computation.

## 4 PROPOSED METHOD

In this section, we describe our proposed method, GADGET. We first describe how to decompose $N$ records with a potentially complicated distributions into $K$ Gaussian-like distributions. Using the surrogate probability density derived from the Gaussian decomposition, we design training strategy, which enhance the sampling quality/diversity in a non-trivial way.

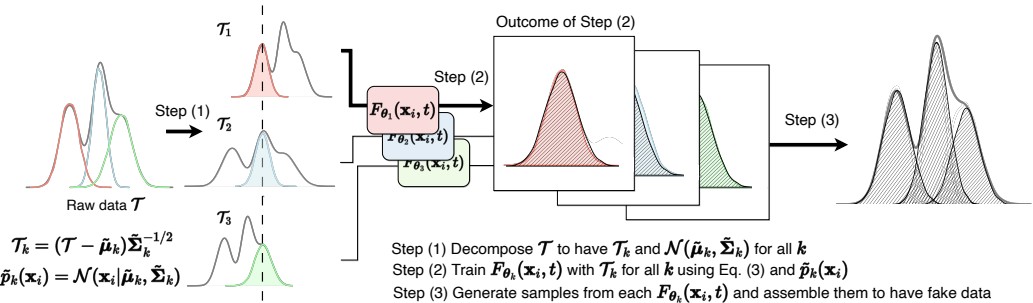

Figure 3: Overall workflow of GADGET. Each color represents each Gaussian-like distribution $\mathcal{N}_k$, and the black hatched distribution means a distribution learned by $k$-th model $F_{\boldsymbol{\theta}_k}$.

### 4.1 GAUSSIAN DECOMPOSITION FOR FINDING THE SURROGATE PROBABILITY DENSITY FUNCTION

**Definition 1.** *The ground-truth density function $p(\mathbf{x})$ of tabular data is unknown. However, the following surrogate density can be derived for a record $\mathbf{x}$ for $k$-th Gaussian distribution $\mathcal{N}_k$: $\tilde{p}_k(\mathbf{x}) = \mathcal{N}(\mathbf{x}|\tilde{\boldsymbol{\mu}}_k, \tilde{\boldsymbol{\Sigma}}_k)$, where $\tilde{p}_k(\mathbf{x})$ is a surrogate probability density of $k$-th distribution and $\tilde{\boldsymbol{\mu}}_k$ and $\tilde{\boldsymbol{\Sigma}}_k$ are its parameters.*

Gaussian mixture models can be used to represent the probability distribution of a multi-dimensional variable as a weighted sum of multiple multivariate Gaussian distributions. The original tabular data distribution contains a number of variables, and we use Eq. 1 to model the original distribution. The step (1) in Fig. 3 shows an example of the Gaussian mixture distribution decomposition using GMMs, where we find 3 Gaussian distributions.

### 4.2 DENSITY-BASED TRAINING STRATEGY

In this section, we discuss the density-based training strategy in detail.

**Definition 2.** *Let $\mathcal{T}$ be a tabular data consisting of $N$ records to learn. Let $\mathcal{T}_k$ be a set of $N$ records that are i) normalized with $k$-th Gaussian-like distribution's parameters, i.e., $\tilde{\boldsymbol{\mu}}_k$ and $\tilde{\boldsymbol{\Sigma}}_k$, and ii) associated with its surrogate probability density $\tilde{p}_k(\mathbf{x})$. We note that $\mathcal{T}_k$ has all $N$ records — $\mathcal{T}_k$ is not a subset of the original tabular data. One can consider that $\mathcal{T}_k$ is a projection of the original tabular data onto $\mathcal{N}_k$ (cf. Step (1) of Fig. 3).*

In training, $\mathcal{T}k$ is used to train $k$-th model $F\boldsymbol{\theta}k$ after weighting each record $\mathbf{x}$ based on the surrogate density $\tilde{p}k(\mathbf{x})$ in self-paced learning. We note that all records are used in the training to maximize the sampling diversity.

**Definition 3.** *Let $\mathbb{1}_k(\mathbf{x})$ be an indicator variable to denote whether the record has the highest probability density for $\mathcal{N}_k$, i.e., $\tilde{p}_k(\mathbf{x}) = \max_j \tilde{p}_j(\mathbf{x})$.*

#### 4.2.1 SELECTIVE SPL-BASED TRAINING

In our proposed method, we train $K$ model with $\{\mathcal{T}_k\}_{k=1}^K$ separately. We use the same network architecture for each model $F_{\boldsymbol{\theta}_k}$ with different parameters $\boldsymbol{\theta}_k$. To further reduce the training complexity, our training method is based on the self-paced learning (SPL) customized from (Kumar et al., 2010) by us, which controls the training level by weighting the loss function according to the training records' importance or difficulty. We consider a training record with high probability density is of importance. Our proposed objective function $\mathcal{L}_k$ for $F_{\boldsymbol{\theta}_k}$, which aims to learn the overall distribution of $\mathcal{N}_k$, is as follows:

$$\mathcal{L}_k = \frac{1}{N} \sum_{i=1}^{N} v_{k,i} l_i(\boldsymbol{\theta}_k), \tag{6}$$

where $l_i$ is the original loss of a base model, e.g., GANs, SGMs, and flow-based models, for $i$-th training record and $\boldsymbol{\theta}_k$ is the parameters for $k$-th model. $\mathbf{v}_k = [v_{k,1}, v_{k,2}, \ldots, v_{k,N}]$ is an importance vector for SPL:

$$\mathbf{v}_k = \mathbf{v}_{k,scaled} + \frac{1 - \mathbf{v}_{k,scaled}}{10,000} \times epoch. \tag{7}$$

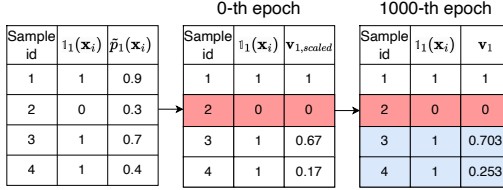

Figure 4: Example of importance vector scheduling during training. Red color indicates that the importance of the cell does not increase throughout the training, while blue color indicates the opposite case.

Fig. 4 is an example of importance vector scheduling of selective SPL-based training of $F_{\boldsymbol{\theta}_1}$. First, we scale $\tilde{p}_1(\mathbf{x}_i)$ for all $i$ to be in $[0, 1]$ by the min-max scaling to have $\mathbf{v}_{1,scaled}$ which contains the initial weights multiplied to training records (cf. Fig. 4 (Middle)). During the SPL training, $v_{1,i}$, for which $\mathbb{1}_1(\mathbf{x}_i) = 1$, is increased linearly w.r.t. training epoch (cf. Fig. 4 (Right)). Note that importance of sample 2 does not increase since it does not belongs to $\mathcal{N}_1$. At the start of the training, the training records with relatively low $\tilde{p}_1(\mathbf{x}_i)$ which belongs to $\mathcal{N}_1$ are hardly trained. After 10,000 epochs, all belonging samples are fully trained because $v_{1,i}$ for all $i$ where $\mathbb{1}_1(\mathbf{x}_i) = 1$ are increased to 1.

The training strategy provides us with the following two benefits: i) By allowing the model to focus on important records at the beginning of the training, it can better learn the key records of the distribution, whose densities are high, in an efficient way. ii) This allows us to train the $k$-th model with a main focus on $\mathcal{N}_k$ while maintaining the information of records that do not belong to $\mathcal{N}_k$ (to increase its sampling diversity). Therefore, we do not lose information about records that lie on the decision boundary where two or more distributions overlap.

### 4.3 TRAINING ALGORITHM

We describe the overall training process in this section along with the training algorithm in Algorithm 1. First, we initialize $\boldsymbol{\theta}_k$ and decompose the raw data to have $\mathcal{T}_k$ for all $k$. After the decomposition, we can calculate $\tilde{p}_k(\mathbf{x}_i)$ for all $k$ and $i$. Using $\mathbf{v}_{k,scaled}$ calculated with $\tilde{p}_k(\mathbf{x}_i)$, we compute $\mathcal{L}_k$ and update $\boldsymbol{\theta}_k$ for all $k$. Throughout the training, the model focuses on $\mathcal{N}_k$ along with the non-belonging records which have high $\tilde{p}_k(\mathbf{x}_i)$.

The following proposition shows us one more advantage of GADGET, which is we can theoretically estimate the difficulty of training $k$-th model — a deep generative model can be understood as a mapping function from a prior distribution to a target data distribution, and it is well known that a large Wasserstein distance between them means they are disparate distributions and therefore, it is hard to find such a mapping function (Arjovsky et al., 2017).

**Proposition 1.** *Since $k$-th model is trained for $k$-th Gaussian-like distribution, the generation process is reduced to mapping from a Gaussian prior, which is typically a unit Gaussian distribution $\mathcal{N}(\mathbf{0}, \mathbf{I})$, to $k$-th Gaussian-like distribution. The Wasserstein distance between the two Gaussian distributions $d = W_2(\mathcal{N}(\mathbf{0}, \mathbf{I}); \mathcal{N}(\tilde{\boldsymbol{\mu}}_k, \tilde{\boldsymbol{\Sigma}}_k)$ is $d^2 = \|\tilde{\boldsymbol{\mu}}_k\|_2^2 + \mathrm{Tr}(\mathbf{I} + \tilde{\boldsymbol{\Sigma}}_k - 2(\mathbf{I}^{1/2}\tilde{\boldsymbol{\Sigma}}_k\mathbf{I}^{1/2})^{1/2})$.*

---

**Algorithm 1** How to train with our proposed method

---

Initialize $\boldsymbol{\theta}_k$ and process the raw data for $\mathcal{T}_k$, where $k \in \{1, 2, \ldots, K\}$
Calculate $\tilde{p}_k(\mathbf{x}_i)$ for all $k$ and $i$
```
/* Train models based on our
   proposed selective SPL-based
   training                    */
```
1 **for** *each $k$* **do**
2      **for** *each epoch* **do**
3          Update $\mathbf{v}_k$ using $\tilde{p}_k(\mathbf{x}_i)$ with Eq. 7
4          Update $\boldsymbol{\theta}_k$ with Eq. 6
5 **return** $\{\boldsymbol{\theta}_k | k = 1, 2, \ldots, K\}$

---

**How to generate fake tabular data** When we generate fake tabular data, we make the data have the same cardinality of distributions as the original data. We let the $k$-th model generate $\sum_{i=1}^{N} \mathbb{1}_k(\mathbf{x}_i)$ records and merge $k$ subtables to have the same number of records to training data.

## 5 EXPERIMENTS

### 5.1 EXPERIMENTAL ENVIRONMENTS

In this section, we provide our experimental environments. The statistical information of our datasets is in Appendix B.1, and descriptions of baselines are in Appendix B.2.

**Evaluation method** A brief description of our experimental environments is as follows: i) We use 8 baseline methods — for our study, we design a SGM for tabular data in Appendix A — and 11 real-world tabular datasets for classification and regression. ii) In general, we follow the "train on synthetic, test on real (TSTR)" framework (Esteban et al., 2017; Jordon et al., 2019) to evaluate the quality of sampling. Details are in Appendix B.3. For `Identity`, we train them with real training data and test with real testing data, whose score can be a criterion to evaluate the sampling quality of various generative methods in a dataset. iii) We leverage coverage (Naeem et al., 2020) to measure the diversity of the fake data. Coverage is the ratio of real records that have at least one fake record among their neighbors, which are found by the K-NN algorithm. iv) We use various metrics to evaluate in various aspects. We mainly use F1 (resp. $R^2$) for the classification (resp. regression) TSTR evaluation in the main paper.

### 5.2 EXPERIMENTAL RESULTS

We discuss the proposed method in terms of the generative learning trilemma (Xiao et al., 2022). The hyperparameter settings for our experimental results are in Appendix B.4, and full results, including the standard deviation after 5 runs, are in Appendix C.

We show the experimental results where `GADGET` is applied to the existing tabular data synthesis methods, i.e., CTGAN, RNODE and SGM — these are representative GAN, flow, and score-based models for tabular synthesis. The model performance and runtime are summarized in Table 1. CTGAN-GADGET,

Table 1: The existing tabular data synthesis methods trained with and without `GADGET`

| Methods | Sampling Quality | Sampling Diversity | Runtime |
|---|---|---|---|
| CTGAN | 0.3282 | 0.3952 | 0.1417 |
| CTGAN-GADGET | **0.4487** | **0.4352** | **0.0973** |
| RNODE | 0.4701 | 0.5091 | 16.3990 |
| RNODE-GADGET | **0.4802** | **0.5141** | **14.9962** |
| SGM | 0.4736 | 0.5692 | 17.4736 |
| SGM-GADGET | **0.6314** | **0.7137** | **3.5390** |

RNODE-GADGET, and SGM-GADGET mean CTGAN, RNODE, and SGM trained with `GADGET`, respectively. In general, Gaussian decomposition tends to enhance sampling quality and diversity, leading to improved performance in all three `GADGET`-trained methods across almost all datasets (For full results, refer to Appendix C). Additionally, Gaussian decomposition reduces the training complexity of RNODE and CTGAN and significantly decreases the number of learnable parameters in them. As a result, the sampling time can also be considerably reduced.

### 5.3 SGM-GADGET

Since SGM-GADGET exhibits the best performance among the three methods, i.e., CTGAN-GADGET, FFJORD-GADGET, and SGM-GADGET, we compare it with the 8 existing tabular data synthesis methods.

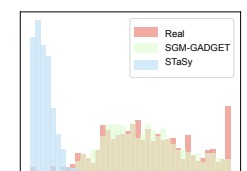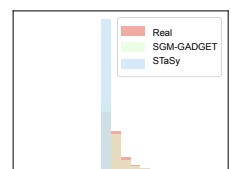

Figure 5: Column-wise histogram of the original data and the fake data by SGM-GADGET and STaSy. (Left) 'cholesterol' from `Heart Disease` and (Right) 'default' from `Bank`.

**Sampling quality** In Table 2, we summarize the sampling quality of the baseline methods and SGM-GADGET. Advanced GAN-based methods, i.e., CTGAN, TableGAN, and OCT-GAN, perform to some degree, whereas MedGAN and VEEGAN show impractical performance. In general, score-based generative models, i.e., STaSy and SGM-GADGET, outperform other generative models in large margins. Especially in `Nursery`, SGM-GADGET show a 27.0%

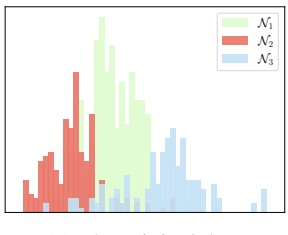

(a) The original data

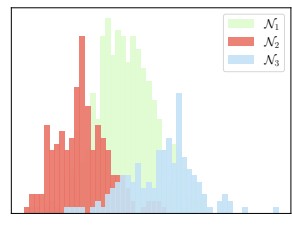

(b) Fake data by SGM-GADGET trained using GMM decomposition

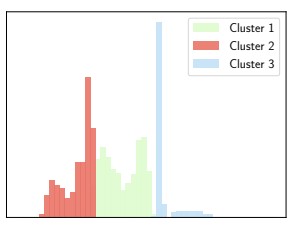

(c) Fake data by SGM-GADGET trained using K-Means clustering

Figure 6: Column-wise distributions of the 'cholesterol' column from `Heart Disease`. (a) Each color represents the histogram of the decomposed real data records using GMM. (b, c) The colors represent the histograms of the fake data records from each SGM, where for (b), the training data is decomposed using GMM, and for (c), the data is decomposed using K-Means.

improvement in F1 score compared to STaSy, with STaSy achieving a score of 0.566 and SGM-GADGET achieving a score of 0.719. In `Absenteeism`, SGM-GADGET exhibit an 943.8% improvement in $R^2$ compared to STaSy, with STaSy achieving a score of 0.016 and GADGET achieving a score of 0.167.

In Fig. 5, we show the column-wise histograms of 2 columns from `Heart Disease` and `Bank`. SGM-GADGET shows much more reliable data distribution than STaSy, showing similar distribution as that of the real data. Fig. 6 (a) and (b) show the column-wise histograms of real and fake data where each Gaussian-like distribution is represented in different colors. The distributions of fake data by SGM-GADGET are similar to those of real data.

**Sampling diversity** In Table 2, we summarize the experimental result in terms of sampling diversity. MedGAN and VEEGAN show poor performance in terms of the sampling diversity, which means they suffer from mode collapse. TableGAN shows reasonable coverage, and score-based generative model, STaSy, shows the best score again among the baseline methods. SGM-GADGET shows the best performance in terms of the sampling diversity again.

Table 2: The generative learning trilemma Xiao et al. (2022). For the sampling quality, we report average values of F1 and AUROC for classification datasets, and average values of $R^2$ and RMSE for regression dataset across the datasets. For the sampling diversity, we report coverage Naeem et al. (2020). We report wall clock time for runtime. The best (resp. the second best) results are highlighted in bold face (reps. with underline).

| Methods | Sampling Quality | Sampling Diversity | Runtime |
|---|---|---|---|
| Identity | 0.6929 | 1.0000 | - |
| MedGAN | -0.2635 | 0.0801 | 0.0223 |
| VEEGAN | -0.5995 | 0.0204 | 0.0219 |
| CTGAN | 0.3282 | 0.3952 | 0.1417 |
| TVAE | 0.2792 | 0.4320 | 0.0201 |
| TableGAN | 0.3743 | 0.5255 | **0.0128** |
| OCT-GAN | 0.3900 | 0.3194 | 1.6101 |
| RNODE | 0.4701 | 0.5091 | 16.3990 |
| STaSy | 0.5966 | 0.6797 | 21.9437 |
| SGM-GADGET | **0.6314** | **0.7137** | 3.5390 |

**Sampling time** SGMs are notorious for their long sampling time, as they require a large number of steps in the reverse process. SGM-GADGET consists of $K$ SGMs, and one might assume that the sampling complexity of SGM-GADGET is of the order $K$. However, it is important to note that SGM-GADGET is trained using *probability flow*, where we generate fake records using the neural ordinary differential equation (NODE) based on $F_{\boldsymbol{\theta}_k}$. By employing an augmented ODE, we are able to generate samples from $K$ SGMs efficiently. The sampling algorithm for STaSy and SGM-GADGET are presented in Appendix D.

In Table 2, we report averaged wall-clock times taken to generate 10,000 records with the methods, which are measured in the same environment. As shown, two early GAN-based methods, i.e., MedGAN and VEEGAN, generate 10,000 samples within about 0.2 seconds, which is a highly fast runtime, and advanced GAN-based methods, i.e., CTGAN and OCT-GAN, take a relatively long but reasonable time. In our experiments, TableGAN shows the fastest sampling speed.

STaSy's sampling time is more than 1,000 times as slow compared to the GANs and VAEs. SGM-GADGET, though, improves the sampling speed by reducing the number of parameters in $F_{\boldsymbol{\theta}_k}, \forall k$.

SGM-GADGET improves the sampling time by 83.9% compared to STaSy. In Appendix E, we provide a summary of the number of learnable parameters for each method. Table 17 shows the number of learnable parameters in SGM-GADGET is significantly decreased compared to STaSy, except for Bank and Heart Disease. This reduction in parameters has resulted in an impressive decrease in sampling time.

### 5.3.1 DATA DECOMPOSITION WITH K-MEANS CLUSTERING

Our data decomposition for tabular data is based on GMMs. K-Means clustering, another popular unsupervised clustering method, can be a substitute for GMM. We provide a comparison between the decomposition methods in this section.

For an importance vector $\mathbf{v}_k$ for K-Means decomposition, we use the distance of each record to the centroid it belongs, where the closer to the centroid records is, the greater weight given. The experimental result is in Table 3. In general, decomposing the data using GMM is better to help the model training in terms of sampling quality and diversity. However, we also observe cases where K-Means decomposition works much better, such as in Absenteeism, Concrete, and Fish.

Table 3: The ablation study on GMM vs. K-Means. For the sampling quality, we report F1 score or $R^2$ (resp. coverage) for the sampling quality (resp. diversity).

| Datasets | Sampling quality | | Sampling diversity | |
|---|---|---|---|---|
| | GMM | K-Means | GMM | K-Means |
| Absenteeism | 0.167 | **0.314** | **0.222** | 0.367 |
| Bank | **0.567** | 0.456 | **0.865** | 0.054 |
| Clave | **0.607** | 0.501 | **0.715** | 0.694 |
| Contraceptive | **0.447** | 0.413 | **0.897** | 0.796 |
| Concrete | **0.835** | 0.831 | 0.552 | **0.820** |
| Customer | **0.362** | 0.334 | **0.745** | 0.399 |
| Fish | 0.548 | **0.557** | **0.945** | 0.828 |
| Heart Disease | **0.876** | 0.827 | **0.895** | 0.634 |
| Nursery | **0.719** | 0.619 | **0.523** | 0.497 |
| Obesity | **0.917** | 0.762 | **0.765** | 0.098 |
| Spambase | **0.901** | 0.878 | **0.727** | 0.077 |

This means these datasets can be divided more effectively based on distance rather than distribution.

In Fig. 6 (b) and (c), we provide the column-wise histograms where each subset is shown in a different color. Compared to Gaussian decomposition, each subset of K-Means decomposition has a proper range but fails to reproduce the desired frequency.

### 5.3.2 SENSITIVITY ANALYSIS ON THE NUMBER OF GAUSSIAN DISTRIBUTIONS

Table 4 shows the sampling quality and diversity with respect to the number of Gaussian-like distributions $K$, which is a hyperparameter in our experiments. SGM-GADGET with $K = 1$ is the same as SGM, which is trained without selective self-paced learning. We discover that the overall performance tends to better when $K$ is small. We conjecture that this is because even though the $\mathcal{N}_k$ is not exactly Gaussian distribution, $K$ with 2 or 3 is enough to reduce the training complexity.

Table 4: The sampling quality and diversity with respect to the number of Gaussian distribution $K$

| Datasets | Sampling quality | | | Sampling diversity | | |
|---|---|---|---|---|---|---|
| | 2 | 3 | 5 | 2 | 3 | 5 |
| Absenteeism | 0.160 | 0.074 | **0.167** | 0.271 | **0.555** | 0.222 |
| Concrete | **0.835** | 0.825 | 0.804 | 0.552 | **0.555** | 0.441 |
| Nursery | 0.671 | 0.717 | **0.719** | 0.518 | **0.528** | 0.523 |
| Spambase | **0.901** | 0.891 | 0.514 | **0.727** | 0.643 | 0.000 |

We also found that the sampling diversity is largely influenced by $K$, and the fake data is prone to be less diverse as $K$ increases. This is because as $K$ increases, $\sum_{i=1}^{N} \mathbb{1}_k(\mathbf{x}_i)$ for each $k$ decreases, letting each model overfitted to each distribution.

## 6 CONCLUSIONS & DISCUSSIONS

We presented a method to decompose the raw data using GMM for density-based model-agnostic training method, i.e., selective SPL-based training, which helps the model to concentrate better on likely data records. The training method also helps to alleviate the training complexity by making each decomposed distribution analogous to the prior distribution. We highlight that by applying the training method, the generation process of the score-based diffusion model, which is known for its large computation, can be simplified and reduce the sampling time while maintaining its sampling quality and diversity.

**Ethic Statement** As tabular data synthesis methods improve their performance, they also raise privacy risks, as they have the potential to disclose sensitive patterns or information about individuals in the original dataset. This presents a challenge for researchers and practitioners who intend to utilize generative models for tasks such as data augmentation, synthetic data generation, or privacy preservation.

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
