# OpenReview forum: "Enhanced Model-agnostic Training of Deep Tabular Generation Models"
_ICLR.cc/2024/Conference — Submitted to ICLR 2024_

### Official Review · Reviewer_wKre · 2023-10-30

**Soundness:** 3 good
**Presentation:** 3 good
**Contribution:** 2 fair
**Rating:** 6
**Confidence:** 3

**Summary:**

This paper presents a novel method  — GADGET — to improve the quality of tabular data synthesizers. The key idea is to learn a GMM over the records so as to produce K datasets whose distributions are easier to learn. One can then train any tabular synthesizer on the simpler datasets, generate data from the K trained synthesizers, and pick the generated data based on the assignment from the GMM. The paper shows GADGET improves synthetic data quality and synthesizer runtime.

**Strengths:**

- The idea of dividing the learning problem into simpler learning problems for tabular synthesis via a GMM is simple, effective, and a generalization that can be applied to existing methods.
- The proposed methods improve data quality and make the training simpler and faster.
- The paper is easy to read, and the Appendices contain information and data that answer almost all my questions.

**Weaknesses:**

- An important component of the GMM is the number of mixtures K: after reading the paper, I am not super clear on the effect of different K.

**Questions:**

Section 2:
- Figure 1: t=0 is not shown, would be good to show the original (the target distribution).
- Figure 2 caption: “each axis means time t.” I don’t quite understand why there are two time axes. Isn’t a comparison between the original distribution (t=0) and the learning-in-progress distributions (t > 0) enough? Also, does top middle panel show that learning made things worse?
- I would expect the Concrete example to be divided into K=5 instead of K=2.

Section 4:
- Is discrete/categorical variables simply treated as numerical variables in the computation of the covariance matrix of the GMM?
- I don’t quite understand the rationale and use of Proposition 1. Of what quantity is N(0,I) the prior? According to the SPL schedule, the weighting based on p_k becomes non-normal pretty quickly, so I am not sure what distance from the prior to the k-the Normal captures. Lastly, is the Wasserstein distance used anywhere?

Section 5:
- Why not use GADGET for all the baseline methods? Would be good to know how GADGET improves existing methods in general.
- Is FFJORD-GADGET the same as RNODE-GADGET?
- The R2 for Absenteeism seems very low despite the nearly 10x improvement from GADGET. Does this have to do with the small data size and the larger number of discrete columns?
- Instead of showing particular examples of distribution match in Figure 5, it might be worth it to just compute resemblance metrics for all the columns (across all the datasets) to quantify the similarity between the real and fake column distributions.

Section 6:
- It is quite surprising that K-means would be better than GMM at all. I am not sure if the distance- vs distribution-based dividing is a satisfying reason. It feels like the GMM should be better because every record has a partial membership in each Gaussian component, whereas this is not the case for K-means. In other words, the GMM is like a probabilistic generalization of k-means. Do the authors have any comments on that?
- There seems to be an optimal K. Table 1 shows that K=1 is not as good as K>1 for sample diversity and other measures. Then the paper states that large K hurts sample diversity because of overfitting. Since K is crucial to the whole idea of the decomposition, I feel the authors could do a more thorough investigation on K. Currently, it’s just a search within the set K={2,3,5}. It might be good to search through a larger range of K, including K=1. I am curious about questions such as: (1) does the optimal K correlate with dataset size? (2) does larger K always improve sampling quality and decrease diversity?

---

### Official Review · Reviewer_gU1v · 2023-10-30

**Soundness:** 1 poor
**Presentation:** 2 fair
**Contribution:** 1 poor
**Rating:** 1
**Confidence:** 3

**Summary:**

The authors consider the problem of generative modeling of tabular data, presenting their method GADGET. This method first decomposes the distribution using a Gaussian Mixture Model to then train a generative model per mixture component.

**Strengths:**

The reviewer does not appreciate any relevant point.

**Weaknesses:**

The idea of structuring tabular data into different Gaussian components follows the right direction, but it is certainly not novel. As a reference, see the paper by Ma et al. 2020: "VAEM: a Deep Generative Model for Heterogeneous Mixed Type Data," which should be included in the experimental settings.

The paper does not read well and lacks soundness in referring to and evaluating the different features of the proposed method. For instance, in section 2, the authors merely use a single example to argue that decomposing into K Gaussians is better. In Section 4 I do not see where exactly the method is explained (just in Figure 3?), and it contains definitions that do not define anything! Check for instance Definition 1.

Empirical results are not conclusive, only one database is included in Table 2. Also, how do you extend this to tabular data containing different data types? how do you tackle missing data?

**Questions:**

Already included in the previous subsection.

---

### Official Review · Reviewer_hPk2 · 2023-10-31

**Soundness:** 3 good
**Presentation:** 1 poor
**Contribution:** 2 fair
**Rating:** 3
**Confidence:** 4

**Summary:**

This paper proposes a new approach to generative modelling of tabular data entitled `GADGET`. This method is based on Gaussianization: first the data is decomposed into a mixture of Gaussians, and then each component of the mixture receives its own generative model. For training each component generative model, the authors propose to use "self-paced learning" that weighs the loss associated to each datapoint by its likelihood of being in that particular mixture component. `GADGET` can be used to augment any generative technique for tabular data, turning the overall modelling problem in $K$ sub-problems which are each ostensibly much easier than the original modelling problem. Various experiments testing the effectiveness of `GADGET` on downstream tasks are performed, including some interesting ablations.

**Strengths:**

I'll list some strengths below in a list so that they can be referenced later as needed. The list is not ordered by importance.

1. Tabular data has generally been underserved by the generative modelling community, despite being the most common type of data observed "in the wild". I appreciate research in this topic overall as I believe it to be an important problem.
2. The use of self-paced learning is quite interesting. I am not very familiar with the topic but it appears that this could perhaps be useful in other domains.
3. The motivation that we want things to be easier Gaussian sub-problems is quite clear to me. The targeted experiments in Section 2 supporting these claims are reasonably convincing.
4. Figure 3 is quite helpful.
5. I found the variety of experiments to be quite cool, especially the ablation in Section 5.3.1 that compared your approach to a related decomposition using K-means clustering.

**Weaknesses:**

Next I'll list some weaknesses. Altogether, I have found that the weaknesses outweighed the strengths. I'll elaborate below (again not necessarily ordered by importance):

1. There is a reason that tabular data has been underserved by the generative modelling community: it is typically hard to do. Several have argued that this is because you cannot easily anticipate the inductive biases of tabular data, unlike say images or natural language, and that successful approaches to -- and architectures for -- discriminative tabular data tasks are not easily ported into the generative domain. A lack of discussion on these points is a major weakness of the paper: it makes it hard to believe that the techniques used in `GADGET` will generalize to a wider range of tabular datasets or work particularly well in the wild, including on higher-dimensional data.
    - Further to this point, it may be worth discussing why this technique has not been used for image-based or language-based modelling, as there seems to be no reason why it wouldn't also apply there.
2. On this same line, I think other important discussions / references are missing:
    - It was recently noted that several advances in tabular data generation are often outperformed by SMOTE (Synthetic Minority Over-sampling Technique), which is a simple interpolation-based sampler for tabular data generation. As such I believe that all generative models for tabular data should compare against SMOTE.
    - If Score-Based Generative models are being used, it would make sense to compare against TabDDPM, as it is the most recent and performant SGM-based generative method for tabular data.
    - It also took me a while to realize the deep link between the Gaussianization piece and score-based modelling, which itself uses Gaussian distributions throughout the modelling process and conceivably works better on Gaussianized data. It is perhaps possible to reframe the paper through this lens, as it may be able to answer some questions about why we would expect the method to perform well.
    - Previously, there have been attempts at incorporating Gaussian mixtures into generative modelling. For example, masked autoregressive flows considered using a Gaussian mixture for the latent space distribution to achieve improved expressiveness. However, discussions related to approaches using Gaussian mixtures in generative modelling are sorely missing from this work.
    - The recently-proposed "union of manifolds hypothesis" also appears relevant here.
3. Another weakness of the related work is that the related work is simply presented, but not at all analyzed. I understand there are space constraints but there are other parts of the paper that can be arguably trimmed down or streamlined to accommodate some improved analysis here.
4. I have several thoughts on the clarity and presentation of the paper, which overall make the paper lack polish. The impression I get from reading the paper is that the work was hastily put-together, and that not enough time was taken to produce quality writing. I'll list my thoughts along these lines below:
    - There are a decently high number of typos throughout the manuscript.
    - The exposition at the end of page 3 feels not only unnecessary, but is also quite confusing. The first paragraph of 3.1 should have been sufficient if you were short on space. The discussion may also have been streamlined by presenting through the lens of conjugate priors.
    -  Starting on Page 6, the discussion in Section 4.2 becomes incredibly confusing. I don't believe $v_{k, scaled}$ was actually introduced anywhere yet was presented in (7). Then, the paragraph immediately following is almost entirely sentences which are unclear. Figure 4 is also not very helpful. This section needs to be overhauled.
    - In Section 5.2, the "Generative Learning Trilemma" is apparently a focal point of the experimental section, but is not actually introduced. It is awkward to have such a fundamental part of the exposition be completely deferred to the cited reference.
    - RNODE and FFJORD appear to be used interchangeably, between 5.2 and 5.3.
5. **There is no limitations section**. This suggests a lack of perspective on how this work sits within the overall literature.
6. It would have been nice to see more _quantitative_ discussions about training time and memory usage in the experimental section, considering you are training and storing $K$ times as many models. It is briefly mentioned that you have fewer learnable parameters, but it would be nice to quantify that.
7. Figure 5 appears to be cherry-picked.
8. This paper is all about making Gaussianized sub-problems for easier generative modelling of tabular data. However, the quality of the Gaussianization was never assessed. It would be nice to see some metric for how well the mixture itself models the data, and relate that to the overall quality of generative model based on downstream metrics.
9. An ablation on self-paced learning would be interesting here.

**Questions:**

1. In Section 2 it is stated that the lower row has a "vector field becomes significantly more intricate than the upper row as $t$ approaches $0$", which is then used to motivate that Gaussianization is a good thing to do. However, as per weakness #8 above, how good is the Gaussianization practically?
2. In Figure 2, why doesn't the diagonal -- especially the $t = 0, 0$ element -- always at maximum covariance?
3. Again with Figure 2, how do we surmise that high similarity of the vector field over time corresponds to higher representation power or better modelling performance?
4. Does Proposition 1 only apply when the data is _actually_ Gaussian? Not just approximately Gaussian? And what is the actual utility of Proposition 1?

---

### Official Review · Reviewer_PxXe · 2023-10-31

**Soundness:** 2 fair
**Presentation:** 2 fair
**Contribution:** 2 fair
**Rating:** 5
**Confidence:** 4

**Summary:**

The paper proposes a new training method called GADGET (Gaussian Decomposition-based Generation of Tabular Data) which can be applied to any generative model for tabular data. The method decomposes the complex distribution of tabular data into a mixture of $K$ Gaussian distributions and then trains one model for each decomposed distribution.  In addition to this, the authors propose a self-paced learning algorithm, whose goal is to focus the training on those data records that have high surrogate probability density.

**Strengths:**

The idea seems nice and the presented experimental results are positive.

**Weaknesses:**

My main worry about the paper is that it seems to have been heavily inspired by the works that presented CTGAN and Stasy, however, it never mentions (nor compares itself to) the shared features of GADGET and the above models. For example, for CTGAN the authors wrote: "CTGAN approximates the discrete variables to the continuous spaces by using Gumbel-softmax". While this is true, one could argue that one of the major contributions of CTGAN is the proposal of modeling the distribution of each continuous feature with a variational Gaussian mixture model. This intuition seems to have heavily influenced this work, however, this is never mentioned. In the same way, the authors of StaSy also propose a self-paced learning algorithm. However, the one presented in this paper is never directly compared to the StaSy one.
Could you please provide some insights on the differences?

Also, could the authors explain why they used 10,000 in the denominator to define $\mathbf{v}_k$? Does the number of epochs need to be exactly 10,000? Why?

How many modes were set for CTGAN? If I remember correctly the default number is 10. Would it make sense to train it why more than 1?

I think an ablation study to check what is the impact of the self-paced learning would be really useful to evaluate the impact of each of the two contributions.

Since GADGET can be applied to any model, it is more interesting to check what is its impact on each model rather than comparing SGM-GADGET with all the standard models (e.g. MedGAN). As SGM on its own does already better than MedGAN, what sense does it make to compare SGM-GADGET with it? It would be much more interesting to compare MedGAN vs MedGAN-GADGET.

Minor comments:
- introduce the abbreviation SGM before using it
- page 4: "Itô"?
- page 5: $\mathcal{T} k$ should be $\mathcal{T}_k$
- page 7: why does SGM-GADGET have the best performance among CTGAN-GADGET, FFJORD-GADGET and SGM-GADGET? What is FFJORD-GADGET?

**Questions:**

See above

---

### Meta-Review · Area_Chair_3MDc · 2023-12-06

**Metareview:**

This paper introduces GADGET as a novel approach for generating tabular data, decomposing its complicated distribution into a mixture of Gaussian distributions, and training individual models for each decomposed distribution with the aid of the proposed self-paced learning algorithm. While reviewers acknowledged the merits of this research, particularly expressing interest in the motivation and the self-paced learning algorithm, they also identified significant drawbacks, such as issues related to presentation clarity and a lack of comparative analysis with existing studies and so many.

Regrettably, the authors failed to address any of the reviewers' concerns or queries, leaving unresolved issues and unaddressed questions. The absence of responses from the authors prevented the resolution of concerns raised by the reviewers, ultimately leading to the decision to reject the paper.

**Justification For Why Not Higher Score:**

the authors failed to address any of the reviewers' concerns or queries, leaving unresolved issues and unaddressed questions.

**Justification For Why Not Lower Score:**

n/a

---

### Decision · Program_Chairs · 2024-01-16

Reject